# Religious Education in Poland during the COVID-19 Pandemic from the Perspective of Religion Teachers of the Silesian Voivodeship

Roman Buchta [1], Wojciech Cichosz [2,*] and Anna Zellma [3]

1 Faculty of Theology, University of Silesia, 40-007 Katowice, Poland; roman.buchta@us.edu.pl
2 Faculty of Theology, Nicolaus Copernicus University, 87-100 Toruń, Poland
3 Faculty of Theology, University of Warmia and Mazury, 10-719 Olsztyn, Poland; anna.zellma@uwm.edu.pl
* Correspondence: cichosz@umk.pl

**Abstract:** The COVID-19 pandemic has influenced all spheres of life. It has an impact on the education of children and youth. The authors' research focused on religious education during the pandemic by the Roman Catholic Church in Poland in the Śląskie Voivodeship. The criterion for choosing the environment was dictated by demographic conditions. The Śląskie Voivodeship has the highest population density per square kilometer, which contributed to the largest number of virus infections. The principal purpose of the research was to gather religion teachers' opinions concerning organization and implementation of the remote teaching of religion in the Silesian Voivodeship (Województwo Śląskie). So far, there has been no research conducted in the field of the abovementioned issues in the area chosen by the authors. The authors' research, carried out from June to August 2020, covered 700 people, which accounts for 18.7% of the religious education teachers working in the Śląskie Voivodeship. The results of the authors' own research allowed formulating a conclusion that the COVID-19 pandemic contributed to the development of modern information and communication competencies of all participants of religious education. Entities involved in this type of education recognized and used multiple opportunities offered by information and communication technologies, which can be seen in the respondents' declarations describing various forms of their didactic and educational on-line work. The COVID-19 pandemic has motivated parents to become more involved in the religious education of younger school-age children. The surveyed religion teachers declared that, thanks to the activity and the help of their parents, the children systematically participated in religion lessons and carried out orders and educational tasks without major problems. Thus, indirectly, parents of children of a younger school age were covered by religious education and were subject to pedagogy. Nevertheless, as teachers have pointed out, the virtual world makes it impossible to form authentic interpersonal relationships. The research confirmed the thesis, according to which religious education carried out in distance learning limits the complete implementation of its objectives, since an upbringing in faith calls for community that necessitates direct contact between the pupil and the teacher.

**Keywords:** school; religious education; religion teacher; COVID-19 pandemic; Silesian Voivodeship

## 1. Introduction

The SARS-CoV-2 virus pandemic that causes COVID-19, which has lasted since November 2019, has become a fundamental variable shaping various dimensions of human existence in the modern world. It has a clear and significant impact on all spheres of individual and social life worldwide, including in Poland. Researchers recognize that education is one of the areas of social life that has been hit hardest by the COVID-19 pandemic (Czetwertyńska 2020; Jankowiak and Jaskulska 2020). The need to organize and conduct distance learning has been a new challenge for teachers. Research conducted by Centrum Cyfrowe has shown that before the outbreak of the pandemic, 85.4% of the

teachers surveyed had no previous experience with remote teaching (online classes), yet 48% had no major difficulties with using digital tools (Centrum Cyfrowe 2020). The effects of the COVID-19 pandemic are also observed in the area of systematic religious education of children and youth at Polish schools, conducted by various churches and religious associations in Poland (Szpet and Staniś-Rzepka 2020).

The authors' research focused on the religious education of children and adolescents conducted during the pandemic by the Roman Catholic Church in Poland, in the Śląskie Voivodeship. The criterion for choosing the environment was dictated by demographic and social conditions. The Śląskie Voivodeship has the highest population density per square kilometer in Poland (Runge 2014). This, as research shows, contributes to the number of SARS-CoV-2 virus infections (Gumułka 2020). Since the beginning of the pandemic, the largest outbreak has been recorded in the Upper Silesia region. Taking into account the conditions enumerated above, research on religious education in Poland during the COVID-19 pandemic was undertaken from the perspective of teachers of religion in the Silesian Voivodeship.

In this article, religious education is understood, in accordance with the terminology adopted in Polish theological literature, as various didactic and educational activities aimed at supporting Polish students in the holistic development and formation of religious attitudes (Chałupniak 2012; Zellma et al. 2021). Religious education in Poland is carried out mainly in schools and is of a confessional nature. All churches and religious associations have the right to organize it. The organization of religious education in Polish schools is regulated both by the documents of churches and religious associations, as well as state normative acts (Bałoniak 2020). The confessional nature of religious education in Poland is expressed in subordinating it to individual churches or religious associations in terms of the content of teaching and upbringing, didactic aids, and the selection and formation of religion teachers. It is not ecumenical in nature, but it assumes ecumenical education, including shaping the attitude of dialogue, tolerance, and openness (Zellma et al. 2021). The church develops and approves curricula, textbooks, and teaching materials for religious education at school, and submits them to educational authorities for information. The church also influences the employment and dismissal of religious teachers who are required to have a confessional theological preparation. However, the confessional views of religious education in Poland differ significantly. Individual churches and religious associations perform in various extent the functions of teaching, education, and initiation, and correlate them with the didactic, educational, preventive, and caring activities of the school. These differences are also related to the quality of evangelistic activities. In practice, religious education in Polish schools is mainly focused on conveying messages, developing skills, and shaping attitudes.

Religious education in Poland conducted by the Roman Catholic Church is carried out at the level of primary and secondary schools (MEN 1992, 2014). Parents or students themselves (after reaching the age of majority) may choose (in the current legal state) religion, ethics, or neither of these subjects. Therefore, religious education is optional and is offered in the form of religion lessons for two hours a week. At the request of the parents or legal guardians of the child (up to 18 years of age), all public and private Catholic schools are obliged to organize religious education. Religious education conducted by the Roman Catholic Church is correlated with catechesis, preparing for the sacraments in the parish (Zellma and Czupryński 2020). It is unified in many respects.

Religious education provided by the Roman Catholic Church in the Polish school system is an ecclesial activity. It takes the form of educational processes aimed at achieving the goals of Christian teaching and upbringing. It is connected with all the activities of the church, especially the various forms of evangelization (Zellma et al. 2021). As far as possible, it is to be an intellectual deepening and preparation for initiation catechesis conducted in the parish. In religious education in Polish schools, the tasks set by the church are implemented, as well as the selected goals and tasks of the school. The first focus is on Christian education and the teaching of faith. Their aim is also to assist students

in human and religious formation. The second category of tasks carried out in religious education is related to the inclusion of religion teachers in the scope of responsibility for the implementation of the tasks set out in the school's statute, as well as in its educational and preventive programs. A teacher of religion is obliged to actively participate in the education of children and young people in accordance with Christian values. Their tasks also include the correlation of the content of religious education with the content of general education. Religion lessons are subject to pedagogical supervision. In terms of methodology, teachers of religion are assessed by the school management, and in terms of content by diocesan inspectors. Importantly, they are obliged to choose from the goals and tasks of the school those that carry basic ethical and moral values, and to engage in the process of developing the students' critical and creative abilities in order to take responsibility for them and prepare them for active involvement in the life of a democratic society.

Religious education in Poland is carried out mainly at school and aims to make participants of religion lessons more mature and conscious of themselves as individuals, rooted in Polish culture, and capable of creative criticism and reflective affirmation of the social and cultural reality that surrounds them (Kostorz 2018; Łabendowicz 2019; Zellma 2017). Religious education in Polish schools is aimed not only at acquiring new knowledge, but also at developing skills and shaping the attitudes of participants in religious lessons (Bałoniak 2020; Mendyk 2007; Zellma 2020). "Its task is to lead a person to an independent and rational interpretation of human existence and to help them understand themselves and the world in the context of the matter of their religion" (Konferencja Episkopatu Polski 2018).

This study aimed to show various organizational and methodological aspects of religious education in Poland during the COVID-19 pandemic, in connection with the main assumptions of religious education. Taking into account the abovementioned goal, on the basis of the results of the empirical research conducted, the ways of implementing religious education in an online format were shown, with particular emphasis on its values and shortcomings and the role in the implementation of didactic and educational tasks, and its correlation with parish pastoral care. The analyses presented in the article focused on the most important empirical data obtained from religion teachers from the Silesian Voivodeship. On this basis, conclusions were drawn regarding religious education in Poland during the COVID-19 pandemic.

## 2. Research Problem and Hypothesis

This study on religious education in Poland during the COVID-19 pandemic, from the perspective of religion teachers in the Silesian Voivodeship, necessitated the identification of problems and the definition of hypotheses that determined the purpose and scope of the research (Łobocki 2001; Siuda and Wasylczyk 2018). Therefore, it was necessary to specify the research problem by asking questions regarding the essence of the research undertaken (Łobocki 2001; Muszyński 1970). They are related to the uniqueness of the universal, global struggle with the pandemic and the transition of most countries to the system of remote work, including Poland, which students endured for the first time in history. Guided by the need to isolate the research problem, the following questions were asked: Is the implementation of religious education of children and youth fully possible in the remote education environment as a result of the COVID-19 pandemic? What conditions should be met for remote learning to bring about the expected didactic, educational, and formative effects?

To resolve these issues, it was necessary to formulate more specific questions (Sztumski 2019). The first determinant was the subjects of religious education: pupil, parent, and teacher. In this category, the following questions were asked: Does remote learning affect the attitudes of students towards the subject of religion, and to what extent? How are students (participating in religious education) affected by the lack of direct contact with the teacher and peers (positively or negatively), and what development directions should be expected (regression or progress)? To what extent do the religious attitudes of the parents influence the process of their children's religious education, and can par-

ents be expected to become more involved in the process of their religious development? Will distance learning and the use of information and communication technology deepen (strengthen) the student-parent-teacher relationship?

The next category of questions were issues related to teaching tools, means and techniques (Cichosz 2000). A question worth asking is: Do teachers have the appropriate tools, resources, and techniques to conduct an effective teaching and upbringing process in the age of a pandemic, since this form of teaching has been implemented ad hoc (without preparation)? To what extent does the COVID-19 pandemic affect the creativity of religious educators and the willingness to establish closer relationships with students? What tools and forms of teaching are effective in remote work, based on experience to date? What ICT tools should be used for this form of work with a student?

Another category of questions concerned didactics: Does distance learning affect students' involvement in a positive or negative sense? Can we assume that disparities will grow larger between students actively involved in the didactic and educational process, and students with educational difficulties?

Since religious formation is an extremely important factor in teaching religion, the following questions arise: Is the specificity of religious lessons and religious formation helpful in developing religious attitudes, in the absence of direct contact between the student and the teacher, and in the inability to actively and directly participate in the sacraments (e.g., confession, Holy Communion)? To what extent does the involvement of parents in the process of didactic development affect the formation of children's faith in the perspective of difficulties resulting from distance learning? Is distance learning conducive to the religious development of students and if so, what distance education techniques are particularly useful?

The formulation of the research problem, contained in the main question posed above and the detailed questions extracted from it, allowed for the formulation of a research hypothesis (Łobocki 2001) intended to develop a presumed answer to the question contained in the research problem. The hypothesis should concern the relationships taking place in a given domain of reality, the regularities that guide it, and the functional mechanisms of the phenomena under study or their essential properties (Cichosz 2020). In the conducted research, due to the uniqueness of the situation, this problem was determined by numerous questions, allowing for the formulation of the following hypothesis: religious education carried out in distance learning limits the complete implementation of its objectives, since an upbringing in faith calls for community that necessitates direct contact between the pupil and the teacher.

### 3. Organization and the Course of Research

The research concept was first developed bearing in mind the remote implementation of religious education in Poland resulting from the COVID-19 pandemic. It was decided to conduct quantitative research using our own survey questionnaire. It was conducted among religion teachers in the Silesian Voivodeship. The main aim of the research was to collect the opinions of religion teachers on the organization and implementation of distance-teaching religion in the Śląskie Voivodeship. It was assumed that the results of empirical research would allow us to diagnose the situation of remote religious education in the Śląskie Voivodeship, and to draw conclusions based on those results.

The Śląskie Voivodeship is an administrative and territorial division of Poland with an area of 12,333,000 square meters, which is 3.9% of the country's area. It covers the eastern part of Upper Silesia and the western part of Małopolska, including Zagłębie Dąbrowskie, Zagłębie Krakowskie, Żywiecczyzna, and Częstochowa. The seat of the voivodeship authorities is Katowice. The Śląskie Voivodeship is inhabited by about 4.517 million people, which is 11.77% of the entire population of Poland (Główny Urząd Statystyczny 2020). The area covered by the research is characterized by the highest degree of urbanization in Poland, with a population density of 366 people/km$^2$ (the average density in Poland is currently 124 people/km$^2$). The central area of the voivodship is the most densely

populated, with a population of over 1000 people per km$^2$. The urban population constitutes 76.6% of the total population (compared to an average of 60% in Poland), and 23.4% in the countryside (average of 40% in Poland). The main branches of the economy are services and industry. Located in the central part of the voivodeship, the Upper Silesian Industrial District (GOP) is the most heavily industrialized area in Poland. The Rybnik Coal District (ROW) located in the southwest boasts similar data. The Śląskie Voivodeship is also characterized by a significant share of national minorities. According to the 2011 census data, 20,000 Germans live here, together with the Czech and Moravian minorities. The social character of the region is shaped by a significant group of people declaring the Silesian nationality or belonging to the Silesian ethnic group. In the 2011 census, there were 318,000 exclusively Silesian declarations (approx. 7% of the total population in the Śląskie Voivodeship) and 370,000 with double Silesian-Polish identification. In 2019, 3,875,771 people declared their affiliation to the Roman Catholic Church, and 48,871 people declared they were members of the Evangelical-Augsburg Church. The number of preschool and school children (aged 3–18) in the Śląskie Voivodeship in 2019 was 685,040 (Urząd Statystyczny w Katowicach 2020). According to the information collected by diocesan entities responsible for religious education, 546,210 students participated in school religion classes in 2020 in the area covered by the research, which constitutes 85.2% of all children and adolescents. Among them, 426,771 (78%) were children attending kindergartens and primary schools, and 119,439 (22%) were youth attending secondary schools (Borda et al. 2021).

The research was carried out in two stages. First, the survey questionnaire was developed covering various aspects of the organization and implementation of religious education in schools using remote education methods and techniques. In the first half of June 2020, a pilot study was carried out among a group of 30 religious education teachers employed in various types of schools (primary and secondary) in Katowice. This research made it possible to verify the original survey questionnaire. It showed that the questions were properly developed and did not require correction. Therefore, the results obtained in the pilot survey were also included in the final analysis.

In the next stage, which took place after the end of the 2020/2021 school year in the summer months—from June to August 2020—the actual research was carried out. At this stage of the research, religious education teachers were asked to give their opinions on the organization and implementation of religious education during the COVID-19 pandemic, teaching in five dioceses located in the Silesian Voivodeship: Bielsko-Żywiec, Częstochowa, Gliwice, Katowice, and Sosnowiec. In cooperation with the entities responsible in dioceses for the religious education of children and youth in schools, religion teachers were sent a link to the questionnaire form containing nine questions (yes/no or multiple choice) and one open question, asking for personal comments and observations on the issues studied.

The CAWI technique was selected as the survey research technique, i.e., an online survey based on a questionnaire on the World Wide Web, to which respondents had access via a web browser (Zellma 2019). It was used to carry out quantitative research. The respondents were asked to complete an online questionnaire (the so-called self-filling form) (Zellma 2019). In terms of methodology, it met all the criteria that apply when creating a traditional questionnaire. It is an adaptation of a traditional questionnaire to the medium of the Internet. In order to carry out the measurement, the research tool was placed in the Google environment (Google Forms). The technique has many advantages. It is one of the most popular techniques used in modern online quantitative research (Zellma 2019). Thanks to the system, researchers have easy access to a specific target group. In turn, respondents can quickly complete the survey questionnaire. Thus, the online survey allows for quick collection of empirical material. It also ensures full anonymity of the respondents, allows quickly previewing the test results (including access to periodic reports), and provides accessible analysis and presentation of results. "Respondents have more time to ponder upon their responses (...). Placing the questionnaire on the website ensures the standardization of research, as the instructions and the content of the questions as well as the possible rotation of questions are displayed in the same way (...) Its obvious

advantage is the possibility of setting the transition to the next question in such a way that it does not allow the omission of questions and the feeling of greater anonymity and greater freedom in statements" (Zellma 2019, p. 219). Research using an Internet questionnaire allows for targeting a specific selection of respondents. This was very important in our case. It resulted from the clearly defined subject of the planned empirical research. At the first stage of the research, in the conceptual phase, taking into account demographic factors and the development of the COVID-19 pandemic, we assumed that the research would cover teachers of religious education from the Śląskie Voivodeship. At the same time, we were aware that research using an online questionnaire, apart from the unquestionable advantages mentioned above, also has its drawbacks. These include, among others, problems with recruiting respondents. We only reached religious education teachers who used the Internet and were able to use it to answer the questions in the questionnaire. However, we assumed that the SARS-CoV-2 pandemic and the related implementation of educational tasks using distance learning methods and techniques forced all religious education teachers to use the Internet. We were also aware that there was a risk of accidental and multiple filling out of the questionnaire by one respondent, as well as technical problems (e.g., issues with access to the Internet, displaying the questionnaire in a form that made its completion impossible). We made every effort to ensure that all methodological rules were complied with. During the COVID-19 pandemic, there was no other way to conduct empirical research. Besides, there is no perfect research method. Each research method has its strengths and weaknesses (Buchta 2006). However, more and more often, social and theological sciences recognize more advantages than disadvantages of Internet research using a questionnaire. This type of research is the basis for further in-depth qualitative analysis, which can be helpful in verifying theory and improving practice.

Ultimately, in the actual research, 700 people responded positively to the invitation of the organizers to creatively participate in the anonymous questionnaire, which constitutes 18.7% of the 3746 religious education teachers working in the five indicated dioceses. The research results—having the character of an introduction to further scientific analysis—were presented at the national forum in connection with the publication of the pastoral program of the Catholic Church in Poland for 2020/2021 (Buchta et al. 2020).

The analysis of the collected material showed that the most numerous group of respondents were lay women (46.2%), then priests and religious education teachers (27.6%), lay men teaching religion (20.5%), and the least numerous group was nuns (5.6%). Among the surveyed teachers, as many as 62% indicated a city as their working environment, 15.9% the small town environment, and only 22.1% the rural environment. The obtained percentage distribution fully reflects the structure of residence in the Śląskie Voivodeship, which allows us to consider the obtained results to be representative. For the credibility of the research, it is not without significance that access to the Internet is easier in highly urbanized areas (Główny Urząd Statystyczny 2019). The age criterion included in the survey—directly related to professional experience—is faithfully reflected in the age structure of all religious education teachers working in the Śląskie Voivodeship. People from 25 to 35 years old constitute 19.7% (21.5% of respondents), from 36 to 45 years old 31.9% (33.2% of respondents), from 46 to 55 years old 31.9% (31.2% of the respondents), and over 56 years of age 16.5% (14.1% of the respondents). With regard to the educational stage, 76.9% of the surveyed teachers indicated primary school as their workplace, while 23.1% indicated secondary schools (all types). The percentage distribution obtained in the study is faithfully reflected in the number of students participating in religious education at the level of primary (78%) and secondary (22%) schools, according to the latest information collected by diocesan departments of religious education (Borda et al. 2021).

The determinant of the first criterion was a question about the subject of religious education: teachers, students, and parents. When answering the question about the impact of distance learning on students' attitudes towards the subject of religion, and thus on the effectiveness of educational activities, the vast majority of respondents indicated

differentiated attitudes (64.2%). The choice of this option seems to confirm the high degree of difficulty, or even impossibility, of an unambiguous assessment of distance learning. This is also evidenced by numerous individual statements provided by educators. The analysis of all the obtained results shows an even distribution of distance learning effectiveness assessments. A slight improvement in involvement in the majority of students was indicated by 12.3% of the respondents, and a slight deterioration was indicated by 13.4% of the respondents. The extreme judgements are also in balance, as a significant improvement was indicated by 3.6% of teachers, and a significant deterioration in student activity was indicated by 6.3% of teachers. When assessing the impact of the limitations of the pandemic period in terms of the subjectivity of religious education, which requires the creation and maintenance of a living relationship between the teacher and the student, as many as 55.1% of the study participants indicated the loneliness felt by students (56.1%) as the most serious deficit of distance learning and the teacher-student relationship (55.1%). The existing situation, according to 32.2% of teachers, obliges them to try to establish more individual contact with students. At the same time, as many as 50.2% of respondents noticed in this period a greater interest of parents in teaching children and the possibility of establishing closer cooperation with parents thanks to information and communication tools (32.9%). Opinions of teachers pointing to the necessity to involve the parents of children too much (33%) are important for the assessment of the implementation of religious education in a pandemic.

Another category of questions concerned issues related to teaching tools, resources, and techniques. When answering the question about how to conduct religious education lessons during distance learning, the most numerous group of respondents indicated communicating with students via instant messaging in real time (34.9%), a slightly smaller group of teachers described their way of conducting lessons as preparing and providing materials for online use at any time for students (30.7%), and 33.8% of respondents declared sending specific tasks, links, and content for individual processing by students (offline). Taking advantage of the possibility of free expression in the open question provided for in the survey, many teachers declared the simultaneous use of each of the above forms of teaching, which was confirmed by the percentage distribution of the answers obtained. Among the tools used, the most popular were the teachers' own materials (recorded videos, stories, riddles, etc.), which were used by 63.7% of respondents, as well as electronic textbooks and materials for teaching religious education (57.4%). Multimedia platforms (YouTube, dominikanie.pl, Holyweek, etc.) proved to be helpful for 57.7% of teachers. The use of various types of instant messaging (Zoom, Skype, Hangouts, etc.) was appreciated by 53% of the survey participants, while the platforms supporting education (e.g., LearningApps, Kahoot, Librus, Padlet, Teams, etc.) were used in remote teaching by 46.6% of educators. Social media platforms (Facebook, Instagram, TikTok, etc.), used by 24.1% of respondents, were less popular. When asked about the benefits of remote education, the participants of the survey indicated the need to be more creative than before the pandemic during the preparation of lessons (67.7%), the necessary element of which was to expand the existing knowledge and skills in the field of information and communication technologies (65.9%). Only a few teachers indicated difficulties in mastering remote teaching tools (13.1%) and in selecting and applying appropriate teaching methods (12.3%). Distance education was perceived as an opportunity for personal development and for searching for new pedagogical and pastoral initiatives (35.1%). It is significant that, despite many difficulties related to the organization of ad hoc distance teaching of religious education, only 1.1% of the survey participants did not perceive any positive aspects of this form of education.

The didactic questions were intended to obtain opinions on how e-learning affects teachers and students. Fatigue caused by the excessive amount of time spent in front of the computer was highlighted among the most serious threats (68%). This was due to both remotely conducted teaching and the time necessary to develop one's own teaching materials (68%). In the first weeks of the pandemic, a serious problem was the lack of

materials for remote work and difficulties in obtaining them (17.6%). Limitations related to access to a computer (a number of brothers and sisters of school age, parents working from home) had a significant impact on the participation and involvement of students in remote religious education, as indicated by exactly 50% of respondents, and the difficulties of students in mastering tools enabling remote education (24%). In a few institutions, the problem turned out to be the marginalization of religious education by the school management (8.4%).

The analysis of the collected opinions of religious education teachers leads to a clear conclusion that remote teaching exacerbates the disproportions between students actively involved in the didactic and educational process and students with educational difficulties. This applies in particular to religious education provided in special education institutions.

## 4. Conclusions and Discussion

The results of our research fully confirmed the adopted hypothesis, according to which religious education conducted in a remote system limits the full implementation of its tasks. Religious education, in a special way, requires a community that is realized in direct contact between the pupil and the educator. It is difficult to compare the obtained results with the data presented in the literature. There is no analogous research carried out among teachers of religious education. The exception is the study by Szpet and Staniś-Rzepka (2020).

In this context, a question concerning shaping the right attitudes emerges. Does the specificity of religious education lessons, religious formation, in the absence of direct contact between the student and the teacher and the inability to actively and directly participate in the sacraments (confession, Holy Communion), favor the development of religious attitudes? Based on the teachers' observations, it can be concluded that the lack of direct contact between the student and the teacher negatively affects the development of religious attitudes. Common prayer meetings and reflections on the content provided in the core curriculum proposed (organized) for students do not compensate for the lack of active and direct participation of children and young people in the sacramental life (confession, Holy Communion) and are only a substitute for the path of formation of religious attitudes appropriate for the church, especially education for community life and education for participation in the liturgy. Despite a definitely negative assessment of the situation, we should emphasize numerous opinions of those teachers who saw in remote teaching of religious education a rare and exceptional—often the only one during a pandemic—possibility of contact with religious content and with a religion teacher perceived as an official representative of the local church community. This seems to be particularly important in the context of regulations on participation in religious gatherings and in the daily life of the church community introduced at that time (Skworc 2020). This is also confirmed by the research conducted by Szpet and Staniś-Rzepka (2020). The authors argued that during the COVID-19 pandemic, remote religious education was often the only place for evangelization (Szpet and Staniś-Rzepka 2020).

The closure of schools and the transition to distance learning surprised all entities involved in the process of religious education. Parents faced a specific test of responsibility for religious education of the young generation. As a result of the existing restrictions, children and young people experienced both the liturgical celebrations of Easter and subsequent Easter Sundays in the community of their immediate family. According to the instructions of the Silesian bishops, the family community was also the place of a direct preparation of children to receive the sacrament of penance and reconciliation and their First Holy Communion. Parents were therefore faced with the need to support the tasks specific to school religious education and the parish catechumenate (Konferencja Episkopatu Polski 2010; Komisja Wychowania Katolickiego Konferencji Episkopatu Polski 2010; Konferencja Episkopatu Polski 2018; Komisja Wychowania Katolickiego Konferencji Episkopatu Polski 2018). However, the question arises as to whether the parents are aware of their tasks and properly prepared to take them up (Buchta 2020).

According to the assessment of the respondents (50.2%), parents showed interest in teaching religion and cooperating with teachers (32.9%), often even engaging in excess of their abilities (33%). Moreover, it happened that their involvement contributed to the activation of some students who, supported by their parents, were more active. In case of the youngest children, parents very often acted as intermediaries between the teacher and the children, which was conducive to establishing closer student-parent-teacher relations. Thanks to their involvement, it was possible for teachers to undertake educational activities. However, this concerned parents of younger students in classes preparing to receive the sacrament of penance and reconciliation and for their First Holy Communion. The involvement of parents of older children and adolescents in religious education is insufficient. In the opinion of the surveyed teachers of religion, parents very often showed a lack of interest in remote teaching of religion. Therefore, it is difficult to expect that such an attitude of parents would provide real support in the process of religious education.

In view of the fairly widespread belief that remote teaching is not conducive to the religious development of students, it is possible to point at various techniques that are particularly useful in religious education. The results of the conducted research indicated a fundamental problem in the field of interpersonal communication while, in the space of knowledge transfer, they clearly showed the enormous potential of the media and various didactic tools available thanks to new information and communication technologies. Szpet and Staniś-Rzepka (2020) came to similar conclusions. The authors drew attention to the potential of digital technologies in teaching religion. On the basis of their research, they proved that direct contact of religious education teacher with student has been replaced by interactive teaching with the use of digital applications and tools. At the same time, they noticed much more educational value in such communication. While pointing to difficulties, they focused on technical and organizational issues (e.g., turned-off webcams and microphones, absenteeism in religious education classes, no access to the Internet, lack of appropriate hardware and software). They considered it important to be able to get to know students from a different perspective, meet them remotely, and communicate using new digital tools. This aspect in the authors' own research looks significantly different (e.g., due to the educational importance of direct contact between the teacher of religious education and the student). The respondents from the Śląskie Voivodeship focused mainly on educational issues. Perhaps this is because the pandemic has affected different areas of religious education and each religious education teacher has different experiences. They are conditioned, inter alia, by the environment in which they live and works, the age of participants in religion lessons, seniority, methodological and educational competences, and gender.

While deficits are clearly visible in the space of sacramental life, in the area of knowledge transfer, the curriculum assumptions of the subject of religion were feasible. However, the respondents noticed the lack of independent work of students, being overwhelmed by the sheer volume of material meant for individual work from other areas and subjects. Effective transfer of religious knowledge included in the core curriculum, as research has shown, is possible because teachers are open to searching for and applying methods and techniques as well as new tools useful in remote education, similarly to other school subjects. This was also confirmed by the results obtained by Szpet and Staniś-Rzepka (2020). Further data are provided by research on the quality of online education in humanities and science: Polish (mother tongue), foreign languages (e.g., English, German, French, and Spanish), mathematics, biology, chemistry, physics, and others (Plebańska et al. 2020). The aforementioned research covered teachers, school principals, and students. In the opinion of humanities and science teachers, despite the initial difficulties (the technological, methodological, and digital competence of teachers) and the lack of organized administrative support, they managed to meet the new challenge in a satisfactory manner. Thanks to cooperation on forums and social groups, they were able to deal relatively quickly with the transfer of educational activities to the digital world and ultimately increase the

value of this form of education. Its effectiveness was assessed as "more than sufficient" (Plebańska et al. 2020, p. 23).

In the opinion of students asked about the experiences related to remote education in various school subjects, as many as 48% of the respondents appreciated its substantive nature, educational capacity (34%), and conduciveness to conversations about life (16%). Respondents from the above-mentioned group of teachers conducting classes in humanities or sciences indicated as a definite difficulty the lack of direct contact with the student (both in terms of teaching and ordinary interpersonal relations), including the possibility of "activating or counteracting the passivity of pupils" (Plebańska et al. 2020, p. 31), pointing to the risk of exclusion and the negative impact of the lack of direct contact with peers. This dimension of remote education was also noticed by students who emphasized that the burden of acquiring knowledge was shifted too much on their shoulders (performing tasks and exercises with the use of textbooks), and accused teachers of a lack of support and difficulty in obtaining help in the event of difficulties in understanding the material. They emphasized the need for opportunities for discussion, performing joint tasks, group work, and teacher support. The students' expectations concerned, inter alia, increasing the attractiveness of educational materials and the digital tools used. The student's digital competences were high, but some of them had significantly difficulty accessing modern technologies. Interestingly, research shows a significant discrepancy in the assessment of the use of educational platforms (students—13.37%; teachers—31%). At the same time, students appreciated the possibility of participating in classes without the need to have to physically go to school, and indicated the ease of access to materials (even many times), the lack of time pressure, and the possibility of individualizing classes (Plebańska et al. 2020). The results obtained by the authors of this article also confirmed that some students showed increased activity in the lessons and gained the ability to learn independently. On the other hand, it was often chaotic work, with a multitude of various tasks to be performed, accompanied by fatigue and frustration. Hence, they clearly appreciated direct contact with the teacher and support in performing tasks, supervision over the course of learning, and discipline. They unanimously emphasized the lack of contact with peers and educators, and the few opportunities to implement projects developing cognitive and social competences (Matos et al. 2021; Seryczyńska et al. 2021). The students, like the teachers, indicated the need to talk about the content of everyday life, but the research showed that the expectations of students in this regard were clearly greater than that of their tutors (Plebańska et al. 2020).

The problems most often raised by students, as shown by Buchner, Szeniawska, and Wierzbicka, the implementers of another study on distance education during the pandemic, were the lack of technical facilities, insufficient digital competences of teachers, the lack of independent work skills bringing measurable knowledge, and the insufficient interpersonal contact. The so-called "vanishing students" were absent from lessons, either due to the lack of equipment, access to an Internet connection, simple neglect of compulsory schooling, laziness or mental problems, or lack of parental supervision (Buchner and Wierzbicka 2020).

In practice, religious education in Polish schools is oriented towards transferring and assimilating knowledge and shaping Christian attitudes. It places emphasis on carefully selected and communicated truths of faith and strengthening attitudes resulting from the Christian faith. As a result, there are tensions and limitations in religious education in Polish schools with educational and preventive processes. The emphasis on teaching the truths of faith and shaping Christian attitudes often limits the scope of educational activities of teachers of religion. Only to a limited extent does it serve for introduction into the community of the church and conscious participation in the sacramental life. Hence, it requires links with other evangelizing and pastoral activities of the church in the parish community.

Before the COVID-19 pandemic, religious education in Polish schools was provided on-site, similarly to other subjects. Attention is also paid to the correlation of didactic and educational activities at school with parish catechesis. During the COVID-19 pandemic,

due to the epidemic limitations related to the number of people who could participate in the liturgy, parish meetings, and compliance with basic sanitary rules, the abandonment of various forms of parish catechesis was noticeable. This definitely influenced the sacramental life of children and young people. There was no possibility of establishing and maintaining direct contacts, conducting dialogue, being together, joint participation in the liturgy, and thus experiencing the community of the church. The entire educational, upbringing, and evangelizing activity of the Roman Catholic Church was transferred to the virtual space and carried out online.

In view of the above, it should be noted that religious education implemented (understood in Poland) as a "special form of catechesis" aims to teach, educate, and introduce children and young people into the community of the church. For this reason, program documents for teaching religion assume not only the transmission of messages but also the formation of Christian attitudes, such as common prayer and participation in the liturgy. In educational practice, this is expressed through the organization and participation of students in the so-called "school masses", retreats, and services that were suspended during the pandemic. As in the humanities, in which we can include religious education, and in the natural sciences, it is possible to transmit content in line with the core curriculum remotely, but as the surveyed population indicates, in their opinion, there were difficulties in implementing the last element of religious education (education of the community).

In the light of the answers to the open-ended questions provided by religious education teachers, it is easy to see that the strengths of technical possibilities are indicated, while the weakest link in religious formation remains the issue of interpersonal relations, both in the horizontal and vertical (sacramental) sphere, as religious education is a subject that requires a personal student-master relationship in a special way. This is due, inter alia, to the essential assumptions of religious education in Poland, which have a clear educational dimension. It is designed, inter alia, to support children and youth in shaping social attitudes, including developing the ability to share good, listen, accept, and respect.

The authors' own research clearly confirmed the complexity of didactic and information processes in the area of religious education, and undoubtedly indicated not only the great value of direct contact in the didactic and educational process, but also the opening of new perspectives and overcoming the fear of using modern technological innovations. Various techniques of remote work (especially synchronous work) facilitated contact between people, and some parents were able to use this value in establishing closer cooperation with the school, directly participating in the didactic and educational process. Nevertheless, as teachers have repeatedly pointed out, the virtual world makes it impossible to form authentic interpersonal relationships. They noticed that conducting lessons "to avatars" was not conducive to establishing the same dialogue that is possible during classroom teaching. However, the correct human-human interactions are indisputably reflected in the creation of the desired human-God relationship.

The results of the authors' own research allows for the formulation of a final conclusion that the COVID-19 pandemic contributed to the development of modern information and communication technologies and the recognition of their possibilities by all entities present in the educational space. In their responses, religious education teachers postulated the need to create a database of educational materials, textbooks, and other didactic materials in electronic form. The experience of many months of remote education also sensitized them to the need to create safe tools for working online. Religious education teachers participating in the research appreciated the potential of educational platforms and the tools they propose, which they will undoubtedly also use in teaching religious education in schools after the pandemic has ended.

The COVID-19 pandemic has shown that teachers of religious education are open to new information and communication technologies. Under the new educational conditions, they were able to fulfill their didactic and educational tasks. They used new digital tools and their possibilities. They showed creativity. They proved that they could fulfill their due mission in all conditions, including those that are unfavorable for the educational process.

An observation springs to mind that the remote teaching introduced without preparation (overnight) has put both students and teachers in the position of facing a completely "new" challenge. Perhaps this experience will contribute to establishing in-depth relations between them. There is no doubt that pandemic-time experience will be verified in the near future. It should be expected that further in-depth research in this field or comparative research in the space of various school subjects will be undertaken. A new look at the analyzed issues would also make it possible to assess religious education during the COVID-19 pandemic from the perspective of returning to full-time education.

**Author Contributions:** Conceptualization, R.B., W.C. and A.Z.; methodology, R.B., W.C. and A.Z.; formal analysis, R.B., W.C. and A.Z.; investigation, R.B., W.C. and A.Z.; resources, R.B., W.C. and A.Z.; writing, R.B., W.C. and A.Z.; supervision, R.B., W.C. and A.Z.; funding acquisition, R.B. All authors have read and agreed to the published version of the manuscript.

**Funding:** This research was co-financed from the funds granted under the Research Excellence Initiative of the University of Silesia in Katowice (Inicjatywa Doskonałości Badawczej Uniwersytetu Śląskiego w Katowicach).

**Institutional Review Board Statement:** Not applicable.

**Informed Consent Statement:** Informed consent was obtained from all subjects involved in the study.

**Data Availability Statement:** Not applicable.

**Conflicts of Interest:** The authors declare no conflict of interest.

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
