# Peer review of "Religious Education in Poland during the COVID-19 Pandemic from the Perspective of Religion Teachers of the Silesian Voivodeship"

_religions, doi:10.3390/rel12080650_

Round 1

Reviewer 1 Report

The thesis is clearly stated and the methodology is defined and contextualized. There are two main areas for expansion and improvement, however, before this paper is ready for publication:

1) Currently the abstract focuses only on the negative hypothesis that distance learning is not ideal for spiritual education and that the studies bear this out. However, positives, such as increased parent involvement with younger children, should be included alongside the increased technological literacy that occurred (which the authors do adequately include). 

2) The most important area for expansion and contextualization is a section (no more than a page would be necessary) comparing and contrasting the author's findings in comparison with secular education. I.e. the author's conclusions that the removal of sacramental community is a major distinctive negative in distance learning for religious education immediately raises the question: is this quantifiably worse than the impact on secular education? How does it compare with impact on STEM education? How does it compare with the impact on secular humanities education (which arguably shares a lot in common with religious education)? 

I think a section of such contextualization, as a sort of "control group" for the study is essential for the conclusions drawn to have a strong contribution to scholarship. Otherwise it may simply be reinforcing that, like all or most forms of education, religious education too is not optimal when done in a distance format. It needs to be clearer why spiritual education is particularly harmed by the lack of sacramental community. 

Author Response

We would like to thank you so much for your thorough analysis of the text and for pointing out the areas that needed some additional information. We acknowledge each comment. We have addressed them in the attached text. 

Thus, in the discussion and arguments, an extensive (3.5 pages) text has appeared on distance education in the field of humanities and science taught in schools. It has been correlated with religious education in Poland. We trust that citing other studies makes the discussion more in depth. We have clarified the understanding and implementation of religious education in Polish schools.

Addressing the issue of literature, we wish to explain that, so far, the issue of religious education in Poland during the COVID-19 pandemic has not been thoroughly researched. Nevertheless, we supplemented the literature by analyzing the studied issues with the latest available studies on Poland and other countries. In the discussion, we supplemented the reference to the literature in which research on education within general education subjects (e.g., Polish language, mathematics, biology, chemistry, and physics) is analyzed. 

We would like to thank the reviewer for a valuable hint regarding the summary. We supplemented its content with positive aspects.

The answer to the question why religious education has suffered in a special way is explained in the supplement to the article. It should now be more visible as we have shown the understanding, implementation, and purpose of religious education that is present in Polish education. In view of the above, it should be noted that religious education implemented (understood in Poland) as a "special form of catechesis" aims to teach, educate, and introduce children and young people into the community of the Church. For this reason, program documents for teaching religion assume not only the transmission of messages but also the formation of Christian attitudes, such as common prayer or participation in the liturgy. In educational practice, this is expressed through the organization and participation of students in the so-called "school masses", retreats and services that were suspended during the pandemic. As in the humanities, which include religious education, history, social studies, Polish, foreign languages (e.g., English, German, and French), and philosophy, and in the natural sciences (e.g., biology, physics, and chemistry, and mathematics), it is possible to transfer the content in line with the core curriculum remotely, but as the surveyed population shows, there were difficulties in implementing the last element of religious education (community education). Of course, this theme found its place in the content of the article.

Thank you very much. Please accept our compliments.

Reviewer 2 Report

Dear Author(s), 

This is an interesting topic, particularly from the perspective/assumption that religious education necessitates (more so than other subjects) in-person human interaction.  As a reader, I found myself wanting a lot more context about religious education upfront- is the religious education curriculum national, standardized, or is it determined by individual dioceses? Is the curriculum designed to instill Catholicism, or is it ecumenical in nature?  What weight is religion given in the over all curriculum, as compared to say math and language arts?  Do students take religion each year? Are there independent religious schools in Poland or are most schools public?  These are the kinds of questions I was asking upfront which would help me better contextualize the survey and findings.  On that note, I think it would aid the reader for you to foreground the methods and findings with the assumption you flesh out in the conclusion- that the very nature of religious education necessitates or is designed for in-person instruction (because students practice the sacraments, for example).  What does religious education typically look like in a nonCovid context?  

In terms of the methods section, I think you have some information that is unnecessary- like what a hypothesis is supposed to do, why a google survey is appropriate, etc. 

In terms of the findings and conclusions, a table or tables of results would be helpful.  One conclusion I questioned about is how exactly "it can be concluded that the lack of direct contact between the student and the teacher negatively affects the development of religious attitudes".  I feel like that conclusion is a stretch based on the findings reported particularly because you didn't use student metrics.  Perhaps if the instrument was supplied as an appendix or you provided more explanation, I could be convinced.  Or maybe you explain that teachers perceived remote negatively impacted the development of religious attitudes if you asked that in the survey.  I also didn't see in the findings section where it was reported that participants noticed a lack of independent work, as you explain in the conclusion. 

So all of this is to say, I think revising this article to more comprehensively contextualize religious education upfront in Poland and to more fully substantiate the conclusions would strengthen it a lot. 

Author Response

We would like to thank you from the bottoms of our hearts for your thorough analysis of the text and for pointing out the areas that needed some additional information. We acknowledge each comment. We have addressed them in the attached text. 

Responding to the first point, we would like to note that, in the article, in line with the suggestions of the honorable reviewer, we defined the issue of contextualization rather broadly. We trust that this will allow us to understand the issues analyzed by us, regardless of the cultural and social context (the educational system in Poland and the organization of religious education). In the article, we devoted 1.5 pages to this issue.

We understand the reviewer's position regarding the redundant provision concerning e.g. Google survey. In our opinion, it can be removed from the abstract; however, it would be good if it could remain in the body of the article. We believe that leaving this notation may contribute to the dissemination of knowledge to a wide range of readers. Not everyone conducts empirical research. Moreover, various methods, techniques, and research tools can be used for conducting empirical research. 

Taking a stance on the postulate to enrich the article with tables, charts, and summaries, we explain that in the methodology adopted by us we assumed that we will present the results of our own research in a descriptive form (e.g., having regard for the length of the text). We admit that we collected them in tabular and graphical form (charts); however, they constituted the material necessary for us to perform a detailed analysis. We are still convinced that their inclusion in the content of the article is not necessary, and the presented detailed data analysis in a descriptive form is sufficient.

Responding to the last objection, we would like to inform you that we have included new materials (3.5 pages) in the field of humanities and natural sciences in the discussion. In addition, we deepened our understanding of the nature, implementation, and goals of religious education in Poland. This allows to place the studied issues in the context of the school, and at the same time show its uniqueness. Thanks to the procedure, the conclusion should be clearer. We have shown the understanding, implementation and purpose of religious education present in Polish education. In view of the above, it should be noted that religious education implemented (understood in Poland) as a "special form of catechesis" aims to teach, educate, and introduce children and young people into the community of the Church. For this reason, program documents for teaching religion assume not only the transmission of messages but also the formation of Christian attitudes, such as common prayer or participation in the liturgy. In educational practice, this is expressed through the organization and participation of students in the so-called "school masses", retreats and services that were suspended during the pandemic. As in the humanities, which include religious education, and in the natural sciences, it is possible to transmit the curriculum content online, but as the surveyed population indicates, there were difficulties in implementing the last element of religious education (community education). Of course, this theme found its place in the content of the article.

Thank you very much. Please accept our compliments.

Round 2

Reviewer 2 Report

Dear Authors, I sincerely appreciate the additional contextual information you provided to both frame the study and its findings.  It greatly strengthened the article. One quick note: there are a few paragraphs repeated (at least in the version I looked at) on pages 9 and 11.  Lines 454-505 (where the addition was made) are repeated verbatim in lines 530-581 that should be addressed. 

Author Response

Dear Reviewer,

Thank you for your comments and suggestions. Duplicate paragraphs have already been removed. We apologize for the inconvenience caused by technical reasons.

Yours sincerely
